# Evaluation of Liposome-Encapsulated Vancomycin Against Methicillin-Resistant *Staphylococcus aureus*

**DOI:** 10.3390/biomedicines13020378

**Published:** 2025-02-06

**Authors:** Enkhtaivan Erdene, Odonchimeg Munkhjargal, Galindev Batnasan, Enkhjargal Dorjbal, Baatarkhuu Oidov, Ariunsanaa Byambaa

**Affiliations:** 1Department of Microbiology, Infection Prevention and Control, School of Biomedicine, Mongolian National University of Medical Sciences, Ulaanbaatar 14200, Mongolia; enkhtaivan.e@etugen.edu.mn; 2Department of Biomedicine, Etugen University, Ulaanbaatar 14200, Mongolia; 3Mongolian Academy of Science, Institute of Chemistry and Chemical Technology, Ulaanbaatar 14200, Mongolia; 4Experimental Animal Center, Institute of Biomedicine, Mongolian National University of Medical Sciences, Ulaanbaatar 14200, Mongolia; 5Department of Pharmaceutical Chemistry and Pharmacognosy, School of Pharmacy, Mongolian National University of Medical Sciences, Ulaanbaatar 14200, Mongolia; 6Department of Infectious Diseases, School of Medicine, Mongolian National University of Medical Sciences, Ulaanbaatar 14200, Mongolia

**Keywords:** liposome, vancomycin, MRSA, nanocarriers, wound healing

## Abstract

**Background/Objectives:** Methicillin-resistant *Staphylococcus aureus* (MRSA) is a global health concern due to its resistance to conventional antibiotics. This study evaluated the efficacy of liposome-encapsulated vancomycin against MRSA using phospholipids extracted from egg yolk. Liposomes were prepared via the freeze–thaw method, yielding vesicles with an average diameter of 157.01 ± 33.04 nm and a polydispersity index (PDI) of 0.0442, indicating uniformity and stability. Antibacterial activity was assessed using the microdilution method. Liposome-encapsulated vancomycin demonstrated complete bacterial growth inhibition (100%) against MRSA ATCC 2758 at dilutions of 10^1^ and 10^2^, compared to only 50% inhibition by free vancomycin at 10^1^. At higher dilutions (10^3^), liposome-encapsulated vancomycin maintained 70% inhibition, whereas free vancomycin was ineffective. In vivo studies using a murine wound infection model revealed that wounds treated with liposome-encapsulated vancomycin achieved superior healing, with complete tissue regeneration observed by day 14. Histological analysis showed reduced inflammation and enhanced tissue recovery in liposome-encapsulated vancomycin-treated groups, compared to fibrosis and persistent necrosis in free vancomycin-treated groups. By enabling sustained drug release and improved bioavailability, liposomal formulations minimized required dosages and systemic toxicity, reducing the risk of resistance development. This study highlights the clinical potential of liposome-encapsulated vancomycin as a scalable, cost-effective treatment for MRSA, particularly in resource-limited settings.

## 1. Introduction

The natural environment allows humans to coexist with microorganisms, yet bacterial infections significantly contribute to the global rise in infectious diseases. Bacterial infections are a leading cause of morbidity and mortality worldwide with mortality from infectious diseases including community-acquired pneumonia (CAP) at 2.5 million annually, leptospirosis at 58,900, typhoid fever between 75,000 and 208,000, and diarrhea at 1560 [1,2,3,4,5].

High mortality rates from bacterial infections, particularly in premature infants and young children, are closely associated with the spread of bacteria resistant to multiple antibiotics. This resistance is linked to the functional and structural characteristics of bacteria, including mutations and the transfer of resistance genes [6,7,8,9,10]. According to the 2019 report “The Threat of Antibiotic Resistance in the United States”, antibiotic-resistant pathogens cause over 2.8 million infections and more than 35,000 deaths annually. Europe reports 25,000 deaths per year due to antibiotic-resistant infections, while Asia (specifically India) reports 700,000 deaths annually. Projections indicate that, by 2050, antibiotic-resistant bacterial infections will be among the leading causes of death globally [11,12,13,14,15,16,17].

One of the leading causes of death from infectious diseases is *Staphylococcus aureus* (*S. aureus*). *S. aureus* not only persists on environmental surfaces but also colonizes the nasal cavity and skin of humans and animals, facilitating both direct and indirect transmission. MRSA, a common cause of skin infections, is resistant to most antibiotics, complicating treatment [18,19,20]. A 2023 World Health Organization (WHO) study reported *S. aureus* prevalence rates of 22.27% in the Americas, 16.57% in the Western Pacific, and 10.93% in Europe. A study in the Eastern Mediterranean and African regions showed prevalence rates of 8.55% and 9.04%, respectively. Countries with multiple surveys, such as the United States (23.78%), Great Britain (18.66%), and China (18.07%), exhibited the highest prevalence rates.

Research indicates that approximately 33% of people carry *S. aureus* in their noses asymptomatically, with 2% carrying MRSA. While many carriers do not develop serious MRSA infections, significant progress has been made in reducing MRSA bloodstream infections, which decreased by 17.1% annually from 2005 to 2012. However, the decline slowed between 2013 and 2016, with no significant changes noted. Studies have reported a concurrent increase in MRSA, CAP, and a decrease in *Streptococcus pneumoniae* CAP strains following the implementation of pneumococcal vaccines, particularly in cases complicated by pleurisy. Consequently, staphylococcal infections are rising due to the increase in MRSA infections [21,22,23,24].

Beta-lactam antibiotics are commonly used to treat staphylococcal infections, but the emergence of beta-lactamase-resistant *S. aureus* strains poses significant challenges. Antibiotics such as vancomycin, linezolid, daptomycin, tigecycline, and telavancin are frequently used against MRSA in skin, soft tissue, and postoperative infections [25]. Monitoring and analyzing data on antibacterial drug use is crucial for developing and implementing policies and interventions to promote appropriate antibiotic use at the national level [26]. In Mongolia, amoxicillin, amoxicillin/beta-lactamase inhibitor, ampicillin, and doxycycline accounted for 75% of oral antibiotic consumption in 2018 [27]. A study in 13 Southeastern European countries found these drugs dominated antibiotic use [28]. Antibiotics are categorized into three groups based on their mechanism of action: inhibition of protein synthesis, nucleic acid synthesis, and cell wall synthesis [29,30,31].

Resistance to vancomycin is linked to the *Van* gene, which encodes various resistance phenotypes. Vancomycin resistance is classified into several gene clusters based on DNA sequences of homologs of the ligase *Van* gene, which encodes the key enzyme for synthesizing D-alanyl D-lactate or D-alanyl D-serine (D-Ala D-Ser). At least 11 *Van* gene clusters have been identified, corresponding to the *vanA*, *vanB*, *vanD*, *vanF*, *vanI*, *vanM*, *vanC*, *vanE*, *vanG*, *vanL*, *and vanN* phenotypes, conferring vancomycin resistance. Genes encoding D-alanyl D-lactate ligases, such as *vanA*, *vanB*, *vanD*, *vanF*, *vanI*, and *vanM*, typically confer high levels of vancomycin resistance, whereas genes encoding D-alanyl D-serine ligases, including *vanC*, *vanE*, *vanG*, *vanL*, and *vanN*, generally confer lower resistance [32].

Nanotechnology has revolutionized medicine, particularly in diagnosing diseases, improving methods for preventing and treating cancer, hereditary, and acquired diseases, producing new-generation drugs, and creating effective treatments. Since Aleika Benkham introduced this concept in 1965, advancements in liposome extraction and physicochemical methods have facilitated the modeling of biomembrane functions and liposome structures. Liposomes serve as model systems representing cell surface and biomembrane quality, playing a crucial role in experiments involving antibiotics, cancer drugs, and nucleic acids [33].

Nanotechnology has transformed transdermal drug therapy by providing new strategies to overcome the skin’s natural barrier [34]. Liposomes, developed based on nanotechnology, are drug delivery systems consisting of one or more lipid layers, spherical in shape, and surrounded by an aqueous core filled with hydrophilic compounds. The lipophilic layer, filled with hydrophobic compounds, is biocompatible and biodegradable. Nanosized drug carriers such as liposomes, niosomes, and micelles are designed to deliver drugs through the stratum corneum, the outermost layer of the skin, which poses a significant challenge for transdermal infections [35,36].

Many liposomal formulations have been food drug administration-(FDA) approved for clinical use due to their enhanced drug delivery and safety profiles. These include liposomal amphotericin B for fungal infections and lipoplatin preparations for breast cancer. Other formulations have shown consistent success in preclinical and clinical trials [37]. The necessity for active transport mechanisms for drugs to reach their biomolecules is emphasized in numerous studies, highlighting the importance of ongoing research and development in this field.

Liposome-based nano drug delivery systems offer several advantages over other nano-antibacterial particles due to their biocompatibility, biodegradability, and structural similarity to cellular membranes. These systems enhance drug delivery efficiency by ensuring targeted delivery to the infection site, improving drug stability, and reducing systemic toxicity. Unlike other nanoparticles, liposomes provide sustained drug release, prolonged bioavailability, and minimize the risk of adverse effects. Furthermore, liposomes have shown superior capabilities in penetrating biofilms and delivering antibiotics to difficult-to-reach infection sites, making them particularly effective against antibiotic-resistant bacteria such as MRSA. Evaluating the efficacy of liposome-based antibiotic delivery systems in treating skin inflammation is one of the pressing challenges in modern medicine. Liposomal systems enhance the bioavailability of antibiotics and concentrate their effects at the site of inflammation. However, it is essential to validate the system’s effects at the cellular level in addition to assessing clinical outcomes. While liposomal systems improve drug delivery and efficacy, confirming these effects through histological analysis at the tissue and cellular levels is crucial [38,39].

The use of vancomycin in models of skin inflammation offers an effective approach to combating antibiotic resistance. Demonstrating the therapeutic outcomes of this treatment through histological evidence of changes in tissue structure ensures the reliability and accuracy of this study [40]. Without histological analysis, it is impossible to precisely evaluate key cellular-level changes such as inflammatory cell accumulation, tissue regeneration, and vascular alterations. This limitation could negatively impact the credibility of this study’s findings. Therefore, validating the effects of liposome-encapsulation vancomycin therapy through histological analysis and examining its anti-inflammatory mechanisms at the cellular level are essential.

## 2. Materials and Methods

### 2.1. Extraction of Phospholipids

In this study, we used a protocol for isolating phospholipids from egg yolk, which was developed by Gladkowski et al. in 2012. In the protocol, the egg yolk served as the primary raw material for their research experiment. The procedure began with breaking the egg and weighing the yolk. The yolk was then mixed with acetone (Merck, Darmstadt, Germany) and continuously stirred using a magnetic stirrer (IKA, North Carolina, Germany) for 10 min. The mixture was transferred to a glass flask and centrifuged (Eppendorf, Hamburg, Germany) at 3000 rpm for 10 min. After separation and drying of the precipitate, a mixture of chloroform (Sigma-Aldrich, St. Louis, MO, USA) and ethanol (Merck, Darmstadt, Germany) in a ratio of 2:1 was added, and the mixture was stirred thoroughly for 3 h. The resulting solution was then filtered, and the organic solvent was separated to form a thin layer of lipids. The yield of phospholipids extracted from egg yolk is typically calculated as a percentage of the weight of phospholipids obtained relative to the initial weight of the egg yolk used. The formula for calculating the yield is as follows:(1)$$Yield\;\left(\%\right)=\left(\frac{Weight\;of\;phospholipids\;extracted}{Weight\;of\;egg\;yolk\;used}\right)\times 100$$

### 2.2. Identification of Phospholipids by Infrared Spectroscopy

Infrared spectroscopy was employed to determine the absorption and vibrational spectrum of the phospholipids. The measurements were performed in the wavelength range of 400–4000 cm^−1^ using a tungsten light source and potassium bromide (Sigma-Aldrich, USA) pellets.

### 2.3. Liposome Preparation and Drug Encapsulation

The freeze–thaw method was utilized for liposome preparation and drug encapsulation. A lipid film was prepared using egg–phosphatidylcholine (egg-PC) and cholesterol at a molar ratio of 4:1, with a total lipid content of 4 mg. The lipid films were rehydrated with 1.5 mL of phosphate-buffered saline (PBS, at pH 7.4) (Thermo Fisher Scientific, Mumbai, MA, USA) containing 0.01 g of vancomycin (Sigma-Aldrich, St. Louis, MO, USA). A thin film was formed by connecting a round-bottom flask to a rotary evaporator and subjecting the solvent to 3 cycles of freezing at −20 °C for 20 min and thawing at +45 °C for 10 min. The heated vancomycin solution was added to the thin film, and the mixture was swirled in the round-bottom flask to form monolayer vesicles. After being vortexed for 10 min, the round-bottom flask was removed from the rotary evaporator (Thermo Fisher Scientific, Flawil, Switzerland), and the contents were sonicated for 30 s. Immediately after sonication (Branson Ultrasonics, Brookfield, WI, USA), the dispersion was extruded using a mini-extruder to obtain a monodisperse liposome-encapsulated vancomycin formulation.

### 2.4. Particle Diameter and Polydispersity Index

Particle diameter and polydispersity index (PDI) were evaluated with atomic force microscopy (AFM) (Angstrom Advanced Inc., Stoughton, MA, USA). The final particle diameter was determined using the Stokes–Einstein equation, while the PDI was estimated using Nano diversified technical systems software (version 6.34). All measurements were performed in at least three sets of 10 runs at 25 °C. The polydispersity index was calculated using the following formula:(2)$$\mathrm{Polydispersity index}={\left(\frac{std}{avg}\right)}^{2\;}$$
*std*—standard deviation of the diameter,*avg* is the average diameter where std is the standard deviation of the diameter and avg is the average diameter.

### 2.5. Bacteria Culture Methodology

*S. aureus* ATCC 25923 and MRSA ATCC 2758 were cultured in a normal medium and spread evenly on Baird-Parker (BP) agar (Biolab, Budapest, Hungary), a selective medium. The cultures were incubated at 37 °C for 24 h. Colonies grown on Baird-Parker agar were concluded to be *Staphylococcus aureus*. Two to three separate colonies were selected, and a bacterial suspension to 0.5 McFarland standard was prepared and diluted in 0.9% physiological solution.

### 2.6. Microdilution Method

The microdilution method was performed using a 96-well reagent plate. Each well received 50–100 μL of antibiotic solution and an equal amount of bacterial suspension. For microdilution, six wells were used with Mueller–Hinton broth (Sigma-Aldrich, USA), with the liposome-encapsuled vancomycin solution diluted to final concentrations of 1, 0.5, 0.25, 0.12, 0.06, and 0.03 μg/mL. Additionally, 10 μL of a 10^5^ colony-forming units (CFU)/mL bacterial suspension (*S. aureus* or MRSA) was added to each well. Negative controls included the liposome-encapsuled vancomycin solution without bacteria, and positive controls included Mueller–Hinton broth with bacteria. The test wells contained Mueller–Hinton broth, MRSA, and the liposome-encapsuled vancomycin solution.

### 2.7. Evaluation of Minimal Inhibitory Concentration (MIC)

The minimal inhibitory concentration (MIC) of free vancomycin and liposome-encapsulated vancomycin was evaluated using the clinical laboratory standards institute (CLSI) guidelines. The MIC was calculated as the lowest concentration of antibiotic that completely inhibited bacterial growth after 18–20 h of incubation at 35–37 °C.

### 2.8. High-Performance Liquid Chromatography (HPLC)

HPLC analyses were conducted using an Agilent 1260 Infinity HPLC system (Agilent Technologies, Santa Clara, CA, USA) equipped with a C18 reversed-phase column. The mobile phase consisted of acetonitrile (Merck, Darmstadt, Germany) and water containing 0.1% formic acid (Merck, Darmstadt, Germany). The UV detection wavelength was set at 282 nm, with a flow rate of 1 mL/min and an injection volume of 10 µL. A standard vancomycin solution at 1000 ppm was used for calibration and analysis. The HPLC method was chosen for its precision, sensitivity, and reliability in quantifying vancomycin concentration, retention time, and purity in liposome formulations. This method, adapted and optimized from established protocols, was specifically validated to meet the requirements of this study. Validation parameters included linearity across a concentration range of 0–1000 ppm, demonstrating excellent correlation (R^2^ = 0.9998) on the calibration curve. Free vancomycin was removed using a dialysis membrane MWCO 1000, and the encapsulated vancomycin was quantified using HPLC.

### 2.9. In Vivo Wound Healing Assay

Six- to eight-week-old male C57BL mice were utilized in this study. The mice were individually housed in sterile cages under standard laboratory conditions, including a 12 h light/dark cycle. Sterilized feed and water were provided. Following a one-week acclimatization period, full-thickness dermal wounds were created on the dorsum of each mouse using a 5 mm diameter biopsy punch (Hua Mei, Shanghai, China). The wounds were subsequently inoculated with MRSA at a concentration of 10^8^ CFU.

The survival of all infected mice was monitored daily and meticulously recorded. To assess therapeutic efficacy, the wounds were treated once daily with liposome-encapsulated antibiotics, administered at doses corresponding to the MIC determined in vitro. The experimental three groups were organized.

Throughout the experimental period, body weight and wound size were measured regularly to monitor overall health and treatment efficacy. Wound healing was evaluated by assessing the progression of wound closure and recovery across all groups. On experimental days 1, 3, 6, and 14, tissue samples were collected for detailed histological analysis.

Prior to tissue collection, mice were anesthetized to ensure humane handling. Full-thickness skin samples, including 2–3 mm of adjacent healthy tissue, were excised using a sterile surgical scalpel. Following tissue collection, the mice were humanely euthanized in strict adherence to institutional and ethical guidelines.

### 2.10. H&E Staining

The excised tissues were formalin-fixed, paraffin-embedded, and stained using hematoxylin and eosin (H&E) (Sigma-Aldrich, USA) staining for histological examination. The stained slides were analyzed under a Nikon Ci light microscope (Nikon Corporation, Tokyo, Japan) to evaluate tissue architecture and cellular changes in detail.

### 2.11. Statistical Analysis

The data were analyzed with IBM SSPS version 22.0 software. To evaluate between-group differences, a Chi-square, ANOVA, and *t* test were employed. Statistical significance was determined at a threshold of *p* ≤ 0.05.

## 3. Results

### 3.1. Determination of Phospholipids

The yield of phospholipids extracted from egg yolk is 37.3%. Infrared spectroscopy was utilized to analyze the composition and purity of the isolated phospholipids. The spectrogram obtained, ranging from 400 nm to 4000 nm, confirms the distinct characteristics of the tissue-derived phospholipids.

The spectrogram in Figure 1 presents the infrared absorption characteristics of the phospholipids extracted from egg yolk, spanning wavenumbers from 4000 cm^−1^ to 500 cm^−1^. Key absorption peaks and their corresponding functional groups are identified, providing insights into the composition and purity of the isolated phospholipids. Significant absorption bands are observed in the following regions: O–H Stretching (Decreased Intensity): The intensity of the absorption associated with the hydroxyl (-OH) group is reduced, indicating a lower presence of these groups in the sample. C–H Stretching: Peaks at approximately 2921 cm^−1^ and 2851 cm^−1^ correspond to the stretching vibrations of the C-H bonds in the aliphatic chains of the phospholipids. N(CH_3_)_3_ Group: A prominent absorption band at 1051 cm^−1^ is attributed to the stretching vibrations of the N(CH_3_)_3_ group, which is characteristic of phosphatidylcholine, a major component of egg yolk phospholipids. P=O Stretching: The region between 1226 cm^−1^ and 1051 cm^−1^ exhibits strong absorption due to the P=O stretching vibrations, indicative of the phosphate groups presents in the phospholipids. P–O–C Stretching: An absorption peak at 819 cm^−1^ is assigned to the P–O–C stretching vibrations, confirming the presence of phosphate ester groups.

The spectroscopic analysis demonstrates that the extracted phospholipids retain their structural integrity, as evidenced by the distinct absorption peaks corresponding to key functional groups. The purity of the phospholipids is inferred from the absence of extraneous peaks, indicating minimal contamination or degradation during the extraction process. In summary, the infrared spectrogram provides a detailed molecular fingerprint of the isolated phospholipids, validating the efficacy of the extraction method and confirming the presence of characteristic phospholipid functional groups.

### 3.2. Analysis of Liposome Morphology

Liposome size and uniformity are critical parameters in various applications, including drug delivery systems, nanotechnology, and materials science. This study employed atomic force microscopy (AFM) to determine the height, diameter, and measurement error of liposomes.

Height and measurement error: Table 1 the average height of the liposomes was 43.2183 nm with a measurement error of 6.85 nm. AFM analysis confirmed their uniform size distribution and morphological uniformity.

Analysis of liposomes: The mean diameter was 157.01385 nm with a standard deviation of 33.036 nm. PDI of 0.0442 suggests a narrow size distribution, indicating uniformity and stability.

AFM and HPLC analyses demonstrate that the liposomes have desirable size and stability characteristics, making them suitable for drug delivery and nanotechnology applications (Figure 2).

Encapsulation efficiency was calculated by comparing the initial vancomycin concentration (1000 ppm) with the concentration of encapsulated vancomycin (137.525 ppm) after dialysis. Free vancomycin was removed using a dialysis membrane, and the encapsulated vancomycin was quantified using HPLC. The retention time was 5.342 min, with a UV peak at 200 nm, confirming high purity (986 ppm). The encapsulation efficiency was calculated to be approximately 13.75%, demonstrating effective incorporation of vancomycin into the liposomes. The HPLC method employed for this study proved to be highly effective in measuring vancomycin concentration and purity within liposomal formulations. Its validation parameters, including linearity and selectivity, ensured the accuracy and reliability of the results.

### 3.3. Antibacterial Efficacy of Liposome-Encapsulated Vancomycin

This study evaluated the antibacterial efficacy of liposome-encapsulated vancomycin compared to free vancomycin. It was determined that liposome-encapsulated vancomycin inhibited bacterial growth at doses twice as low as those required for free vancomycin. The results are summarized in Table 2.

Liposome-encapsulated vancomycin demonstrated superior efficacy in inhibiting the growth of both MRSA ATCC 2758 and *S. aureus* ATCC 25923 strains. At dilutions of 10^1^ and 10^2^, liposome-encapsulated vancomycin completely inhibited bacterial growth (100%), whereas free vancomycin achieved only 50% inhibition at 10^1^ dilution and was ineffective at higher dilutions.

For MRSA ATCC 2758, liposome-encapsulated vancomycin maintained 70% growth inhibition at a 10^3^ dilution, while free vancomycin showed no inhibitory effect at this dilution. At a 10^4^ dilution, liposome-encapsulated vancomycin exhibited 10% inhibition, whereas free vancomycin remained ineffective.

For *S. aureus* ATCC 25923, liposome-encapsulated vancomycin consistently inhibited 100% growth up to a 10^3^ dilution and retained 70% inhibition at a 10^4^ dilution. Free vancomycin showed no inhibitory effect beyond the 10^1^ dilutions.

Overall, liposome-encapsulated vancomycin demonstrated significantly higher antibacterial activity compared to free vancomycin, achieving comparable or superior inhibition at lower doses. Statistical analysis confirmed the enhanced efficacy of liposome-encapsulated vancomycin with a *p*-value of 0.02, indicating a significant difference in performance (Figure 3).

These results indicate that vancomycin-encapsulated liposomes have a distinct and more potent inhibitory effect on MRSA and *S. aureus* strains compared to free vancomycin. This suggests that liposomal formulations could enhance the therapeutic efficacy and potentially reduce the required dosage of vancomycin in clinical settings.

### 3.4. Evaluation of Wound Healing Mechanisms in Mice

The wound healing process in mice was evaluated using a standardized full-thickness skin injury model. A 5 mm diameter biopsy punch was used to create wounds on the prepared dorsal surface of each mouse, which were subsequently infected with a bacterial suspension. To prevent contamination, adhesive dressings were applied over the wounds.

The progression of wound healing was assessed on days 1, 3, 6, and 14 post-injuries by monitoring body weight and wound size.

During the study, the treatment group showed an average weight gain of 6 g (22 to 28 g), while the control group gained 3 g (22.7 to 25.7 g). Both groups initially experienced weight loss within the first 24 h post-injury. However, by day 6, the treatment group’s weight gain was observed to be double that of the control group, indicating better recovery and health outcomes. The statistical analysis (*p* = 0.62) suggests no significant difference between the groups, but the trend indicates potential benefits of the treatment (Figure 4).

Throughout the experimental period, the progression of wound healing was evident, suggesting that the therapeutic interventions effectively supported tissue recovery.

Liposomes play a crucial role in stabilizing active pharmaceutical compounds and facilitating targeted delivery to specific sites. Liposomal encapsulation enables localized drug delivery, thereby reducing the required dose and minimizing toxicity. Skin wounds are often susceptible to infections caused by Gram-positive bacteria. Due to their ability to accumulate at wound sites, liposomes provide an effective means to evaluate the targeted delivery of antibiotics. Furthermore, by reducing inflammation and efficiently combating bacterial infections, liposomes have the potential to accelerate wound healing processes (Figure 5).

The histological assessment of skin tissue at various time points following treatment with liposome-encapsulated vancomycin and free vancomycin demonstrates distinct differences in the progression of wound healing. In the liposome-encapsulated vancomycin group (A), mild inflammation and early tissue remodeling were observed on day 1, followed by evident granulation tissue formation with reduced inflammatory infiltration by day 3. By day 6, significant re-epithelialization had occurred, accompanied by the development of well-organized dermal structures. By day 14, complete epithelialization and dermal regeneration were achieved, indicative of accelerated healing with minimal fibrosis.

Conversely, in the free vancomycin group (B), extensive inflammatory cell infiltration and necrotic tissue were present on day 1, and persistent inflammation delayed the formation of granulation tissue by day 3. By day 6, necrosis and purulent inflammation were still evident, further impeding wound repair. Although wound closure was observed by day 14, the healing process was characterized by fibrotic scarring rather than complete tissue regeneration.

A comparative analysis of these findings underscores the superior efficacy of liposome-encapsulated vancomycin, which facilitated faster resolution of inflammation, enhanced tissue regeneration, and improved overall wound healing outcomes. In contrast, the delayed healing and fibrosis observed in the free vancomycin group highlight the limitations of conventional antibiotic delivery in promoting optimal tissue repair. These results emphasize the therapeutic potential of liposomal drug delivery systems in enhancing the efficacy of antibiotics for wound healing applications.

## 4. Discussion

The current study investigated the antibacterial efficacy of liposome-encapsulated vancomycin against MRSA, a significant global public health concern since its emergence in the 1960s. With a reported prevalence of 14.69% among 164,717 individuals across 29 countries, MRSA has remained a formidable pathogen due to its resistance to standard β-lactam antibiotics [41,42]. In the late 1990s, the advent of community-associated MRSA (CA-MRSA) infections further complicated the scenario, as these infections affected healthy individuals without traditional risk factors and displayed high virulence, primarily causing skin and soft tissue infections, necrotizing pneumonia, and necrotizing fasciitis.

Given these challenges, our study focused on evaluating the potential of liposome-encapsulation vancomycin formulations. Liposomes, due to their lipid bilayer structure similar to cellular membranes, offer a unique advantage in antibiotic delivery. They enhance the concentration of antibiotics at the site of infection for extended durations, thereby improving antibacterial efficacy. This study’s methodology involved extracting phospholipids from egg yolk using acetone, ethanol, and chloroform, resulting in a yield of 37.9%. This method was chosen based on previous research by Fahimeh Hajiahmadi et al. (2019), which demonstrated higher encapsulation efficiency of vancomycin using the freeze–thaw method over the ammonium sulfate gradient method [43].

Vancomycin was selected for encapsulation due to its widespread use in treating MRSA infections, particularly in Mongolia. Analytical techniques, including chromatography and UV spectroscopy, confirmed the integrity and purity of the encapsulated vancomycin, with consistent retention times and characteristic absorption peaks at 200 nm and 280 nm [44]. These results ensured the reliability of our encapsulation method and the stability of the antibiotic within the liposome formulation.

Our microdilution studies demonstrated that liposome-encapsulated vancomycin exhibited significantly higher antibacterial activity against MRSA strains compared to free vancomycin [42]. Notably, liposome-encapsulated vancomycin achieved complete growth inhibition at dilutions of 10^1^ and 10^2^, whereas free vancomycin only partially inhibited growth at 10^1^ and was ineffective at higher dilutions. Moreover, liposome-encapsulated vancomycin maintained substantial inhibitory activity at 10^3^ and 10^4^ dilutions, underscoring its enhanced potency.

These findings align with previous research by Abdelkader et al. (2012), Surewaard B. (2016), and Liu Y. (2019), which reported that liposome-encapsulated antibiotics are effective at lower doses than their free counterparts [43,45]. The increased efficacy of liposome-encapsulated vancomycin can be attributed to the sustained release and prolonged bioavailability of the antibiotic at the site of infection, ensuring higher concentrations over longer periods.

This study evaluated the efficacy of liposome-encapsulated vancomycin in promoting wound healing and tissue regeneration, with comparisons to antibiotic-free liposome and untreated control groups. The results highlight the therapeutic advantages of liposome-based antibiotic delivery systems, particularly in combating inflammation and enhancing tissue recovery.

The findings demonstrate that liposome-encapsulated vancomycin significantly accelerated inflammation resolution and facilitated complete tissue regeneration by day 14. This can be attributed to the dual benefits of vancomycin’s potent antimicrobial properties and the sustained release provided by the liposomal formulation. The enhanced bioavailability of vancomycin at the wound site likely maintained effective drug concentrations over an extended period, reducing bacterial load and promoting optimal conditions for tissue recovery. These results align with previous studies reporting the efficacy of liposomal antibiotics in improving therapeutic outcomes [44,45].

From the above results, it can be found that the application of liposome-encapsulated vancomycin observed in this study has significant clinical implications. By reducing the required dosage and increasing the duration of effective drug concentration, liposomal formulations can potentially minimize the risk of resistance development and adverse side effects associated with high-dose antibiotic therapies. Additionally, the ability of liposomes to deliver antibiotics directly to the site of infection may improve treatment outcomes for severe MRSA infections, including those caused by CA-MRSA strains. This efficacy can be attributed to the inherent advantages of liposome-based drug delivery systems, including their ability to encapsulate hydrophilic and hydrophobic drugs, target specific sites, and provide controlled drug release. Compared to other nano-antibacterial particles, liposomes exhibit higher biocompatibility and biodegradability, reducing systemic toxicity and side effects. Furthermore, liposomes’ structural similarity to biological membranes enables improved cellular uptake and better penetration into biofilms, a key challenge in treating MRSA infections. These properties position liposomes as a superior alternative to other nano-delivery systems, particularly in combating multidrug-resistant pathogens

Our study demonstrates the superior efficacy of liposome-encapsulated vancomycin over its free form, showing significant advantages in treating MRSA strains. The enhanced effectiveness is attributed to the improved bioavailability and sustained release of the antibiotic at the target sites, which facilitates higher concentrations at the infection site for extended periods. This results in a stronger therapeutic effect while reducing the required dosage, making it particularly beneficial for treating antibiotic-resistant pathogens like MRSA. Liposomal delivery systems can potentially minimize adverse side effects and decrease the likelihood of resistance development by allowing for lower dosages. The consistent efficacy of liposome-encapsulated vancomycin against various MRSA strains suggests it could be a versatile and reliable treatment option in clinical settings. Furthermore, the successful use of locally sourced phospholipids from egg yolk as a cost-effective alternative to commercially available lecithin highlights an innovative approach that could be especially beneficial in resource-limited settings.

## 5. Conclusions

This study demonstrated the high potential of liposome-encapsulated vancomycin for treating MRSA infections and promoting wound healing. Liposome-encapsulated vancomycin exhibited superior wound healing acceleration, achieving complete tissue regeneration within 14 days. In contrast, free vancomycin resulted in delayed healing and fibrosis. The liposomal delivery system highlighted its advantages, including reducing antibiotic dosage, minimizing side effects, and improving clinical outcomes, thereby confirming its efficacy against antibiotic-resistant bacteria such as MRSA.

### Limitation of This Study

Financial limitations are detrimental to research. The most difficult aspect was using rare and advanced equipment. For example, Zeta (ζ) potential technique and dynamic light scattering (DLS).

## Figures and Tables

**Figure 1 biomedicines-13-00378-f001:**
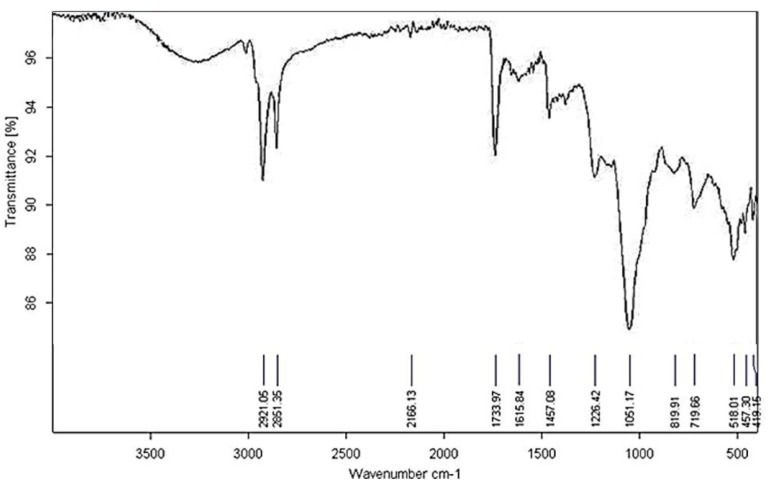
Violet-red spectrogram of phospholipids isolated from egg yolk.

**Figure 2 biomedicines-13-00378-f002:**
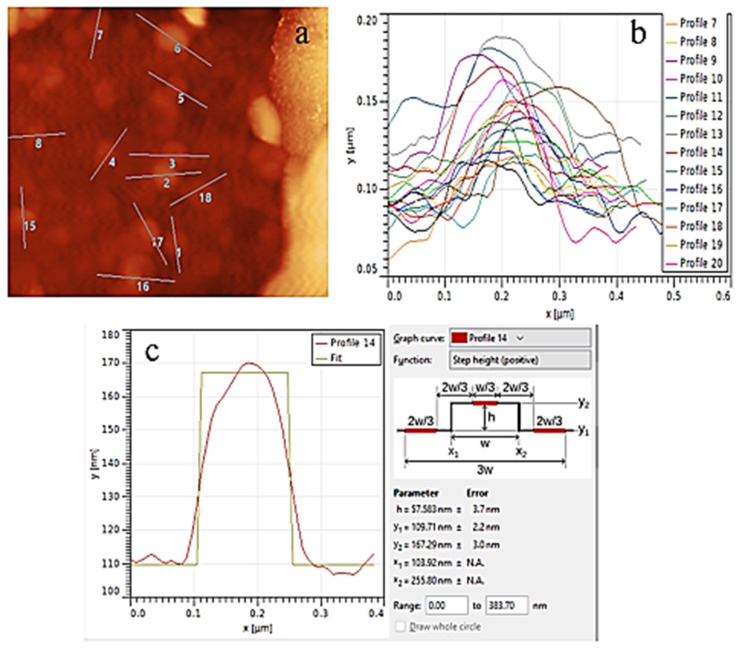
Atomic Force Microscopy (AFM) analysis of liposome morphology. (**a**) AFM Image of liposome slices, where the positions of the slices are marked for further analysis. (**b**) shows the curves corresponding to these slices, (**c**) illustrates an example curve used to define the height and diameter of the liposomes.

**Figure 3 biomedicines-13-00378-f003:**
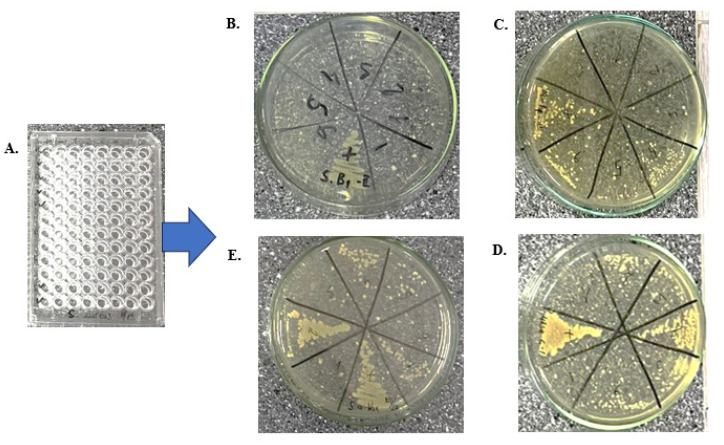
Antibacterial efficacy of liposome-encapsulated vancomycin. Abbreviations: (**A**) microdilution, (**B**) liposome-encapsulated vancomycin against *S. aureus*, (**C**) liposome-encapsulated vancomycin against MRSA, (**D**) free vancomycin against MRSA, (**E**) free vancomycin against *S. aureus*.

**Figure 4 biomedicines-13-00378-f004:**
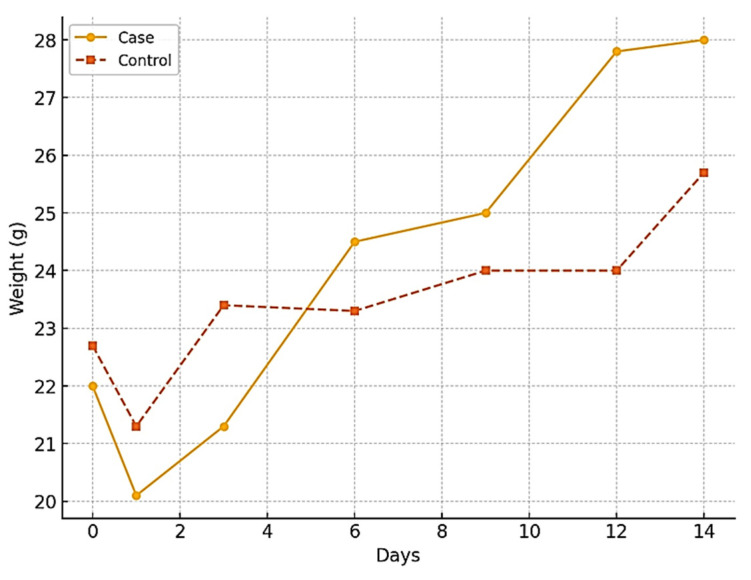
Comparison of the body weight of the experimental groups.

**Figure 5 biomedicines-13-00378-f005:**
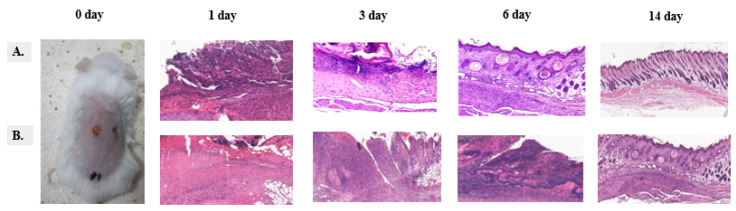
Histological evaluation of skin tissue following treatment with liposome-encapsulated vancomycin and free vancomycin. Staining: H&E, Magnification ×20. Abbreviations: The 0-day image shows the condition of the skin tissue after a 0.5 mm biopsy punch was used to create a wound in both groups before starting the experiment. Skin histological analyses were performed on days 1, 3, 6, and 14 of the experiment. (**A**) liposome-encapsulated vancomycin group, (**B**) free vancomycin group.

**Table 1 biomedicines-13-00378-t001:** Height, measurement error, and physicochemical parameters.

Parameter	Mean ± SD	Error (nm)	PDI
Height (nm)	43.2183	6.85	-
Particle size (nm)	157.01385 ± 33.036	-	0.0442

**Table 2 biomedicines-13-00378-t002:** Bacterial growth inhibition by liposome-encapsulated vancomycin.

Microdilutions	Liposome-Encapsulated Vancomycin		Free Vancomycin	
	MRSA ATCC 2758	*S. aureus* ATCC 25923	MRSA ATCC 2758	*S. aureus* ATCC 25923
	%(n)	%(n)	%(n)	%(n)
$10^{1}$	100 (10)	100 (10)	50 (5)	50 (5)
$10^{2}$	100 (10)	100 (10)	0	0
$10^{3}$	70 (7)	100 (10)	0	0
$10^{4}$	10 (1)	70 (7)	0	0
$10^{5}$	0	30 (3)	0	0
$10^{6}$	0	0	0	0

## Data Availability

The data that support the findings of this study are available on request from the corresponding author.

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
