# Peer review of "Evaluation of Liposome-Encapsulated Vancomycin Against Methicillin-Resistant Staphylococcus aureus"

_biomedicines, 2025, doi:10.3390/biomedicines13020378_

Round 1

Reviewer 1 Report

Comments and Suggestions for Authors

The manuscript entitled "Evaluation of Liposome-Encapsulated Vancomycin Against Methicillin-Resistant Staphylococcus aureus” by Erdene et al. describe the efficacy of Liposome-Encapsulated Vancomycin Against  Methicillin-Resistant Staphylococcus aureus. 

In this study, liposome-encapsulated vancomycin demonstrated bacterial growth inhibition of 100 % against MRSA ATTC 2758. Moreover, in vivo studies using a murine wound infection model showed that wounds treated with liposome-encapsulated vancomycin achieved superior healing, with complete tissue regeneration in 14 days. 

This study is interesting, and the manuscript is written well, there are some suggestions to further improve the quality of this manuscript.

Introduction: The introduction section is written well, the problem statement as well as the research gap are described, and the importance of the current work has been highlighted. 

Materials and methods.

Please mention all the chemicals used in the current work along with their vendors information. Secondly, mention the instruments/ machines used and the names of manufacturers. 

Line # 136 Methodology for the Extraction of Phospholipids: delete this heading from the sentence as it is repeated twice. 

Line # 163 Particle diameter and polydispersity index… complete this sentence. 

Line # 202 revise this heading as “High-Performance Liquid Chromatography (HPLC) analysis” 

Line # 238 Determination of Phospholipids 

The yield of phospholipids extracted from egg yolk is 37.3%. How you calculate the yield? 

FTIR analysis 

Line # 245 Used super subscripts for wavenumber ranger such as 4000 cm-1 throughput this section. 

Line # 287 3.3. Antibacterial Efficacy of Liposome-Encapsulated Vancomycin

Please add a Figure containing the the pictures of plates with bacterial strain to ensure their inhibition. 

Some general comments 

What are the advantages of liposome-based nano drug delivery systems over other nano-antibacterial particles? Please give a brief description.

Please add a display of the antibacterial test results.

Fig. 4 lack scale bars, and the wound looks like a scalpel cut through the skin, unlike a standard full-thickness skin defect model. Also describe what is meant by “A”, “B” 1, 3, 6 and 14 in Fig 4. 

 Overall recommendation: The manuscript could be accepted for publication after major revision.

Author Response

Thank you for your valuable comments and insightful suggestions. We have carefully addressed each of your points and made the necessary revisions to improve the manuscript. To make the changes more visible, we have highlighted the revised sections in the manuscript with yellow. Below is a summary of the revisions:

Comment 1: Please mention all the chemicals used in the current work along with their vendors information. Secondly, mention the instruments/ machines used and the names of manufacturers.

Response 1: We have provided detailed information on all the chemicals used in the study, along with the names of their vendors. Additionally, the instruments and equipment, including the manufacturers, have been specified.

Comment 2: Line # 136 Methodology for the Extraction of Phospholipids: delete this heading from the sentence as it is repeated twice.

Response 2: The repeated heading “Methodology for the Extraction of Phospholipids” in Line #144 has been removed.

Comment 3: Line # 163 Particle diameter and polydispersity index… complete this sentence. 

Response 3: By your comments we have correct. 

Comment 4: Line # 202 revise this heading as “High-Performance Liquid Chromatography (HPLC) analysis” 

Response 4: The heading in Line #215 has been revised to “High-Performance Liquid Chromatography (HPLC) Analysis.”

Comment 5: Line # 238 Determination of Phospholipids The yield of phospholipids extracted from egg yolk is 37.3%. How you calculate the yield? 

Response 5: Thank you for highlighting the need to provide a detailed explanation of how the phospholipid yield was calculated. We have addressed this concern in the methodology section by including the following comprehensive description. Line # 154-157

Comment 6: FTIR analysis  Line # 245 Used super subscripts for wavenumber ranger such as 4000 cm-1 throughput this section. 

Response 6: Wavenumber ranges, such as 4000 cm⁻¹, throughout the FTIR analysis section (Line #266) have been reformatted using superscripts.

Comment 7: Line # 287 3.3. Antibacterial Efficacy of Liposome-Encapsulated Vancomycin Please add a Figure containing the the pictures of plates with bacterial strain to ensure their inhibition.

Response 7: 

Thank you for your valuable suggestion regarding the addition of a figure displaying the bacterial inhibition results to support the findings in the section 3.3 Antibacterial Efficacy of Liposome-Encapsulated Vancomycin (Line #318). In response to this, we have included a new figure showcasing the images of bacterial plates.

The figure provides visual evidence of the inhibition of bacterial strains, clearly demonstrating the antibacterial efficacy of liposome-encapsulated vancomycin compared to controls. The images include:

  • A: Microdilution setup used for initial testing.
  • B, C, D, E: Petri plates illustrating the zones of bacterial inhibition under different conditions (e.g., liposome-encapsulated vancomycin, free vancomycin, and control treatments).

This addition aims to enhance the clarity and impact of the results by visually confirming the findings presented in the manuscript. We hope this addresses your comment effectively. If further modifications are required, please let us know. 

Result of microdilution

Comment 8: What are the advantages of liposome-based nano drug delivery systems over other nano-antibacterial particles? Please give a brief description.

Response 8: The reviewers' suggestions have been accepted, and the necessary additions have been incorporated into the Introduction and Discussion sections.

Comment 9: Please add a display of the antibacterial test results.

Response 9: Thank you for your suggestion to include a display of the antibacterial test results. In this study, we did not use the diffusion method; instead, we evaluated the antibacterial efficacy using the microdilution method to ensure quantitative accuracy. A detailed display of the results, including inhibition percentages for different concentrations of liposome-encapsulated vancomycin and free vancomycin, has been added as part of the response to Comment #7. This addition enhances the clarity of the findings and provides a comprehensive visual representation of the antibacterial activity.

We hope this addresses your concern effectively. Please let us know if any further modifications are required. Thank you for your feedback!

Comments 10: Fig. 4 lack scale bars, and the wound looks like a scalpel cut through the skin, unlike a standard full-thickness skin defect model. Also describe what is meant by “A”, “B” 1, 3, 6 and 14 in Fig 4. 

Response 10: Regarding the wound appearance, the wound was created using a 0.5 mm biopsy punch, not a scalpel. The biopsy punch method was specifically chosen to produce a standardized full-thickness skin defect model. While the edges may appear sharp in the images, this is due to the precision of the biopsy punch rather than a scalpel. This method ensures consistent wound size and depth across experimental groups, meeting the criteria for a full-thickness skin defect model.

We sincerely appreciate your constructive feedback, which has significantly enhanced the quality of our work. Please do not hesitate to let us know if further adjustments are needed.

Thank you once again for your valuable time and insights.

Best regards,
Enkhtaivan Erdene

Reviewer 2 Report

Comments and Suggestions for Authors

Dear Authors

I enjoyed reading your work, which is well-written and contains an interesting perspective on MRSA treatment.

However, I have a few comments:

- HPLC method - you do not write anything about why you use this method.

Did you develop the method, or is it a method from the bibliography? Only in lines 279-280 do you describe that vancomycin was pure.

If you developed it, the work must include information on the method validation along with parameters. Is the method linear in this concentration range at all? Is it selective?

Too many questions. I would need to explain this.

- encapsulation efficiency as determined? How do you know how much drug was enclosed in liposomes? There is no information on whether vancomycin was inside liposomes. 

- what effect do liposomes themselves have on the tested SA strains? You investigated liposomes with vancomycin and vancomycin. What is the impact of pure liposomes without drug? 

This information must be added to the work.

- Figure 3 is unacceptable. The graph must have labeled axes.  In this form, it is inappropriate for scientific work.

- There are punctuation errors at work.

- The bibliography requires adjustment to the requirements of the journal (e.g., unnecessary record of the month of publication)

Author Response

Thank you for your valuable comments and suggestions. We have carefully addressed each of your points and incorporated the necessary revisions into the manuscript. The changes made in response to your comments have been highlighted in blue within the manuscript for clarity. Below is our detailed response to each point:

Comment 1: HPLC method - you do not write anything about why you use this method.

Response 1: The HPLC method, adapted from previously established studies, was deemed the most appropriate analytical technique for this study. Relevant details have been incorporated into the Materials and Methods section.

Comment 2: Did you develop the method, or is it a method from the bibliography? Only in lines 279-280 do you describe that vancomycin was pure. If you developed it, the work must include information on the method validation along with parameters. Is the method linear in this concentration range at all? Is it selective?

Response 2: Thank you for your valuable comment. The HPLC method, adapted from previously established protocols, was optimized and validated for this study. Details regarding the method's validation, including its linearity and selectivity, as well as the encapsulation efficiency calculation, have been incorporated into the Materials and Methods and Results sections. We hope this addresses your concern adequately. Please let us know if further clarification is needed. Thank you again for your feedback!

Comment 3: encapsulation efficiency as determined? How do you know how much drug was enclosed in liposomes? There is no information on whether vancomycin was inside liposomes. 

Response 3: The presence of vancomycin inside the liposomes was confirmed through dialysis and subsequent HPLC analysis. During dialysis, free vancomycin was effectively removed using a dialysis membrane, leaving only the liposomal fraction. The encapsulated vancomycin was analyzed using HPLC, and its retention time and concentration matched those of the calibration standards. These results conclusively demonstrate the successful encapsulation of vancomycin within the liposomes.

Comment 4: what effect do liposomes themselves have on the tested SA strains? You investigated liposomes with vancomycin and vancomycin. What is the impact of pure liposomes without drug?

Response 4: In this study, we included results for vancomycin-loaded liposomes and free vancomycin. The effects of empty liposomes were not assessed, as our previous research, published in the CAJMS journal, demonstrated that empty liposomes exhibited no significant antibacterial activity against Staphylococcus aureus strains. To maintain focus on the novel aspects of this study and avoid redundancy, the effects of empty liposomes were not included here. The prior study’s findings are available for reference in the cited publication.

Comment 5: Figure 3 is unacceptable. The graph must have labeled axes.  In this form, it is inappropriate for scientific work.

Response 5: By your comments we have correct.

Comment 6: There are punctuation errors at work. The bibliography requires adjustment to the requirements of the journal (e.g., unnecessary record of the month of publication)

Response 6:  Punctuation errors will be corrected, and the bibliography will be adjusted to align with the journal's formatting requirements.

We sincerely appreciate your constructive feedback, which has significantly enhanced the quality of our manuscript. If you have any further suggestions or require additional clarifications, please do not hesitate to let us know.

Thank you once again for your valuable time and insights.

Best regards,
Enkhtaivan Erdene

Round 2

Reviewer 1 Report

Comments and Suggestions for Authors

The authors have addressed some of my comments and incorporated the suggested changes in the revised manuscript but the response the comment # 7 has not been aligned with the revised manuscript.  My comment (comment #7) and author's response are appended below

Comment 7: Line # 287 3.3. Antibacterial Efficacy of Liposome-Encapsulated Vancomycin Please add a Figure containing the the pictures of plates with bacterial strain to ensure their inhibition.

Response 7: 

Thank you for your valuable suggestion regarding the addition of a figure displaying the bacterial inhibition results to support the findings in the section 3.3 Antibacterial Efficacy of Liposome-Encapsulated Vancomycin (Line #318). In response to this, we have included a new figure showcasing the images of bacterial plates.

The figure provides visual evidence of the inhibition of bacterial strains, clearly demonstrating the antibacterial efficacy of liposome-encapsulated vancomycin compared to controls. The images include:

  • A: Microdilution setup used for initial testing.
  • B, C, D, E: Petri plates illustrating the zones of bacterial inhibition under different conditions (e.g., liposome-encapsulated vancomycin, free vancomycin, and control treatments).
  • According to authors they have added a new Fig (images of bacterial plats) showcasing the antibacterial activity of of Liposome-Encapsulated Vancomycin but in the revised manuscript I can't see any such Fig. The revised manuscript contain same number of Fig as provided in the old version. 
  • Please add this Fig and properly cite it in the text of revised manuscript in sec 3.3.
  • Moreover, the Figure captions need to be described in a bit more detail so that the author understand what is meant by (A), (B) and 1, 3, 6 and 14 (Fig 4). 

Author Response

Evaluation of Liposome-Encapsulated Vancomycin Against Methicillin-Resistant Staphylococcus aureus

Dear Reviewer 1

Thank you for your valuable comments and insightful suggestions. We have carefully addressed each of your points and made the necessary revisions to improve the manuscript. To make the changes more visible, we have highlighted the revised sections in the manuscript with yellow. Below is a summary of the revisions:

Response to Reviewers’ Comments

Reviewer 1

Comment 1: According to authors they have added a new Fig (images of bacterial plats) showcasing the antibacterial activity of Liposome-Encapsulated Vancomycin but in the revised manuscript I can't see any such Fig. The revised manuscript contains same number of Fig as provided in the old version. Please add this Fig and properly cite it in the text of revised manuscript in sec 3.3.
Response 1: 
By your comments we have correct.

Figure 3. Antibacterial Efficacy of Liposome-Encapsulated Vancomycin

Abbreviations: (А)microdilution, (B)Liposome-Encapsulated Vancomycin against S. aureus, (C)Liposome-Encapsulated Vancomycin against MRSA, (D)Free Vancomycin against MRSA, (E)Free Vancomycin against S. aureus.

Comment 2: Moreover, the Figure captions need to be described in a bit more detail so that the author understand what is meant by (A), (B) and 1, 3, 6 and 14 (Fig 4).
Response 2: 
By your comments we have correct.

Figure 4. Histological Evaluation of Skin Tissue Following Treatment with Liposome-Encapsulated Vancomycin and Free Vancomycin.
Staining: Hematoxylin & Eosin (H&E), Magnification: ×20.

The histological assessment of skin tissue at various time points following treatment with liposome-encapsulated vancomycin and free vancomycin demonstrates distinct differences in the progression of wound healing. In the liposome-encapsulated vancomycin group, mild inflammation and early tissue remodeling were observed on day 1, followed by evident granulation tissue formation with reduced inflammatory infiltration by day 3. By day 6, significant re-epithelialization had occurred, accompanied by the development of well-organized dermal structures. By day 14, complete epithelialization and dermal regeneration were achieved, indicative of accelerated healing with minimal fibrosis.

Conversely, in the free vancomycin group, extensive inflammatory cell infiltration and necrotic tissue were present on day 1, and persistent inflammation delayed the formation of granulation tissue by day 3. By day 6, necrosis and purulent inflammation were still evident, further impeding wound repair. Although wound closure was observed by day 14, the healing process was characterized by fibrotic scarring rather than complete tissue regeneration.

A comparative analysis of these findings underscores the superior efficacy of liposome-encapsulated vancomycin, which facilitated faster resolution of inflammation, enhanced tissue regeneration, and improved overall wound healing outcomes. In contrast, the delayed healing and fibrosis observed in the free vancomycin group highlight the limitations of conventional antibiotic delivery in promoting optimal tissue repair. These results emphasize the therapeutic potential of liposomal drug delivery systems in enhancing the efficacy of antibiotics for wound healing applications.

We sincerely appreciate your constructive feedback, which has significantly enhanced the quality of our work. Please do not hesitate to let us know if further adjustments are needed.

Best regards,
Enkhtaivan Erdene

Reviewer 2 Report

Comments and Suggestions for Authors

Dear Authors

Thank you for your work. Now your manuscript looks better. 

According to Figure 3, please add an axis description.

And please note that after acceptance you should modify the description of manufacture it should be (Merck, Darmstadt, Germany)

Kind regards

Author Response

Biomedicines-3444947

Evaluation of Liposome-Encapsulated Vancomycin Against Methicillin-Resistant Staphylococcus aureus

Dear Reviewer 2

Thank you for your valuable comments and suggestions. We have carefully addressed each of your points and incorporated the necessary revisions into the manuscript. The changes made in response to your comments have been highlighted in blue within the manuscript for clarity. Below is our detailed response to each point:

Comment 1: According to Figure 4, please add an axis description.

Response 1: By your comments we have correct.

Figure 4. Comparison of the body weight of the experimental groups.

Comment 2: And please note that after acceptance you should modify the description of manufacture it should be (Merck, Darmstadt, Germany)
Response 2: 
By your comments we have correct.

We sincerely appreciate your constructive feedback, which has significantly enhanced the quality of our manuscript. If you have any further suggestions or require additional clarifications, please do not hesitate to let us know.

Thank you once again for your valuable time and insights.

Best regards,
Enkhtaivan Erdene